# Bortezomib Pharmacogenetic Biomarkers for the Treatment of Multiple Myeloma: Review and Future Perspectives

**DOI:** 10.3390/jpm13040695

**Published:** 2023-04-20

**Authors:** Antonio Sanz-Solas, Jorge Labrador, Raquel Alcaraz, Beatriz Cuevas, Raquel Vinuesa, María Victoria Cuevas, Miriam Saiz-Rodríguez

**Affiliations:** 1Research Unit, Fundación Burgos por la Investigación de la Salud (FBIS), Hospital Universitario de Burgos, 09006 Burgos, Spain; 2Facultad de Medicina, Campus de Medicina, Universidad Autónoma de Madrid (UAM), 28029 Madrid, Spain; 3Haematology Department, Hospital Universitario de Burgos, 09006 Burgos, Spain; 4Department of Health Sciences, Health Sciences Faculty, University of Burgos, 09001 Burgos, Spain

**Keywords:** multiple myeloma, bortezomib, toxicity, efficacy, pharmacogenetic biomarkers

## Abstract

Multiple myeloma (MM) is a hematological neoplasm for which different chemotherapy treatments are used with several drugs in combination. One of the most frequently used drugs for the treatment of MM is the proteasome inhibitor bortezomib. Patients treated with bortezomib are at increased risk for thrombocytopenia, neutropenia, gastrointestinal toxicities, peripheral neuropathy, infection, and fatigue. This drug is almost entirely metabolized by cytochrome CYP450 isoenzymes and transported by the efflux pump P-glycoprotein. Genes encoding both enzymes and transporters involved in the bortezomib pharmacokinetic pathway are highly polymorphic. The response to bortezomib and the incidence of adverse drug reactions (ADRs) vary among patients, which could be due to interindividual variations in these possible pharmacogenetic biomarkers. In this review, we compiled all pharmacogenetic information relevant to the treatment of MM with bortezomib. In addition, we discuss possible future perspectives and the analysis of potential pharmacogenetic markers that could influence the incidence of ADR and the toxicity of bortezomib. It would be a milestone in the field of targeted therapy for MM to relate potential biomarkers to the various effects of bortezomib on patients.

## 1. Introduction

### 1.1. Multiple Myeloma

Multiple myeloma (MM) is a neoplasm characterized by the proliferation of clonal malignant plasma cells in the bone marrow (BM) that triggers monoclonal immunoglobulin production [1,2]. This disease causes damage to target organs and its symptomatology usually follows the CRAB criteria (hypercalcemia, renal failure, anemia, or bone lesions) [3,4,5].

MM is an adult neoplasm with a median diagnosis age of 69 years old, with its incidence being higher in men [6]. It is the second most common hematological malignancy and causes more than 100,000 deaths per year [7].

Currently, the treatment consists of a combination of different drugs and the choice of treatments depends on the patient’s age, comorbidities, and prognosis. For younger patients without organ dysfunction, the standard treatment comprised induction therapy with a proteasome inhibitor such as bortezomib (BTZ), a immunomodulator (thalidomide or lenalidomide), and dexamethasone, followed by autologous hematopoietic progenitor transplantation with high-dose chemotherapy [8]. Those patients over the age of 65 years with several chronic conditions may be treated with BTZ, melphalan, and prednisolone; lenalidomide and dexamethasone; or melphalan, prednisone, and thalidomide.

### 1.2. Proteasome Inhibitors

The 26S proteasome is a critical complex of the ubiquitin-proteasome system that is localized in the nucleus and cytosol of eukaryotic cells and is responsible for the degradation and regulation of intracellular proteins [9]. The ubiquitin-proteasome system is dysregulated in MM, which triggers increased proteasome activity and thus increased degradation of specific substrates such as the tumor suppressor p53 and inhibition of nuclear factor KB (NF-kB) [10]. There are numerous processes affected by proteasome dysregulation such as cell cycle, oncogenic transformation, and the signaling of proinflammatory cytokine, cell adhesion or anti-apoptotic via BCL2, etc. As a consequence, the MM finds several mechanisms of progression [11].

Some proteasome inhibitors such as BTZ, carfilzomib, and ixazomib have been studied in different assays and target one or more of the 20S proteasome subunits. The mechanisms used by these substances have an impact on the proteasome’s catalytic activity [11].

### 1.3. Bortezomib

BTZ is the first proteasome inhibitor that was successfully used in the treatment of MM. It is an antineoplastic drug whose main mechanism of action is the reversible inhibition of the 20S proteolytic core within the 26S proteasome. As a result, BTZ alters signaling pathways in the cell and BM microenvironment inhibiting cell cycle progression and angiogenesis and, therefore, inducing apoptosis [9,12,13]. The antitumor effect by inhibiting the 26S proteasome prevents the activation of the NF-kB pathway, which is necessary for cell proliferation [14].

BTZ can be administered intravenously and subcutaneously, being systemic exposure similar in both routes. However, subcutaneous administration is associated with greater safety, including a decrease in peripheral neuropathy compared to the intravenous route [13].

The BTZ pharmacokinetic profile is characterized by a two-phase model: the first phase of rapid distribution and the second phase of very long elimination. The distribution half-life is less than 10 min followed by the elimination phase with a half-life of more than 40 h [13,15,16,17]. BTZ is extensively distributed to the peripheral tissues. In the bortezomib concentration range of 0.01 to 1.0 microgram/milliliter, in vitro binding to human plasma proteins averaged 82.9% [18].

## 2. Materials and Methods

A systematic review of the published literature on the implication of polymorphisms in genes of the metabolic pathway and transport of bortezomib that may affect the drug’s effectiveness and safety in MM. Systematic reviews and scientific studies on the subject were also consulted. The selected polymorphisms were chosen based on their frequency in the population and their involvement in enzyme and transporters functionality.

The Preferred Reporting Items for Systematic Reviews and Meta-Analyses (PRISMA) guidelines were followed in conducting this systematic review [15]. These online resources were used to find relevant articles related to “multiple myeloma” AND “polymorphism” OR “SNP” AND “bortezomib” MEDLINE, PharmGKB [16], EU Clinical Trials Register and ClinicalTrials.gov. Similar terms were used across all databases in the search strategy: MM, polymorphism (or SNP, or single nucleotide polymorphism or pharmacogenetics or pharmacogenomics), BTZ and proteasome inhibitor, CYP and ATP binding cassette subfamily B member 1 (ABCB1). The reported review period covered the years from bortezomib’s authorization in 2004 through February 2023. All of the titles and abstracts were separately examined, and each article was assessed by two reviewers (A.S.S. and M.S.R.). When required, disagreements were settled via consensus and in accordance with a third reviewer (J.L.). Studies that satisfied the following criteria were included: (1) studies using pharmacogenetic biomarkers assessed in diagnosed MM patient cohorts in which BTZ was prescribed; and (2) studies reporting efficacy and/or safety variables. After removing duplicates, the systematic search turned up 49 citations, the most significant of which were used in our review.

## 3. Bortezomib Pharmacogenetics

Since its introduction in MM therapy, BTZ has greatly improved patient outcomes. However, like all drugs, BTZ is not free from causing undesired side effects. These adverse drug reactions (ADR) can be nausea, diarrhea, constipation, fever, and peripheral neuropathy [18]. One of the most serious BTZ ADR is peripheral neuropathy, a form of neuropathic pain characterized by severe numbness and paresthesia [19]. BTZ-induced peripheral neuropathy is present in up to 30% of patients and may even lead to treatment discontinuation [20]. Furthermore, despite the fact that the blood-brain barrier keeps BTZ from accessing the central nervous system, it produces neurotoxicity by accumulating in the dorsal root ganglia [21]. Currently, there are no pharmacokinetic markers targeting this neuropathy.

BTZ is metabolized mainly by the isoenzymes complex cytochrome P450 (CYP) [22].

The CYP superfamily of hemoproteins, which shows a species-specific expression pattern, is responsible for the oxidation-based metabolism of both exogenous and endogenous substances [15]. Their location is mainly hepatic although they can be found in other parts of the organism such as the respiratory or intestinal tract [23]. The genes of this superfamily are classified based on the homology of their DNA sequences, thus being divided into families. To date, 57 genes and about the same number of pseudogenes have been described, which are grouped according to their similarity into 18 families and 44 subfamilies (Figure 1) [23].

The CYP1, 2, and 3 isoenzyme families have broad and often overlapping substrate specificities, which generally favor efficient clearance of lipophilic xenobiotics. However, there is some variability in their expression and functions leading to unanticipated drug responses, such as toxicity or unresponsiveness [24]. Each isoform varies in a given population, due to host genetic and non-genetic factors and numerous environmental factors, some of which remain constant throughout life (genotype, sex) while others are dynamic (age, drugs administered, pathologies). Additionally, these variables have varied effects on the various isoforms. For example, CYP2D6 activity is mainly influenced by genetic polymorphisms, whereas CYP3A4 is mainly influenced by sex and the induction or inhibition by a wide range of substances [25].

The isoforms *CYP3A4* (79%), *CYP2C19* (23%), *CYP1A2* (18%), *CYP2D6* (6.6%), and *CYP2C9* (5.4%) have been determined to be primarily responsible for BTZ metabolism [26,27]. Since these enzymes are responsible for the proper metabolism of BTZ, variations in the genes that encode them may result in a direct impact on the metabolism and elimination of BTZ, and hence on the drug’s efficacy and safety profile.

The most significant pharmacogenetic biomarkers related to the pathway on BTZ are described in greater detail below.

### 3.1. CYP3A

CYP3A4, CYP3A5, CYP3A7, and CYP3A43 are the four isoforms that comprise the human CYP3A subfamily, which account for around 30% of the total CYP enzyme content in the human liver [25]. The human CYP3A enzymes CYP3A4 and CYP3A5 are thought to play a key role in drug metabolism [28,29]. In the adult liver and gut, both are particularly prevalent [30].

*CYP3A4* is quantitatively the most important CYP enzyme in adults. With a range of 14.5% to 37% of the hepatic CYP450 pool, CYP3A4 is the most abundant CYP enzyme [31]. In addition, CYP3A4 is the most prevalent enzyme in human intestinal epithelial cells [32]. It contributes to the metabolism of almost 30% of all drugs. Moreover, through first-pass metabolism, intestinal CYP3A4 contributes to the metabolism of several drugs [33]. The PharmVar platform lists 52 alleles within *CYP3A4* (https://www.pharmvar.org/gene/CYP3A4) (accessed on 18 April 2023) [34]. The *CYP3A4**20 (rs67666821) and *CYP3A4**22 (rs35599367) polymorphisms have been associated with a total and partial loss of enzyme function, respectively. *CYP3A4**20 is found at a frequency of less than 0.1% in the European population. However, a frequency of up to 1.2% has been described in the Spanish population, due to a founder effect [35]. The frequency of the *CYP3A4**22 allele is 5% in the European population. Moreover, *CYP3A4**3 was associated with lower LDL-cholesterol plasma levels in hypercholesterolemic patients treated with atorvastatin [36], and the *CYP3A4**2 allele was associated with a reduced intrinsic clearance for nifedipine, midazolam, or testosterone [36]; *CYP3A4**6 causes a frameshift and an early stop codon that might result in decreased CYP3A4 activity [36]. Moreover, subjects carrying the *CYP3A4**18 allele showed higher CYP3A4 catalytic activity [37]. However, the clinical function of *CYP3A4**2, *CYP3A4**3, *CYP3A4**6 and *CYP3A4**18 has not been determined yet [34]; hence, further research is warranted. Regarding BTZ, the study by Zhou et al. found no association between the CYP3A4 metabolizer phenotype and BTZ efficacy or the incidence of peripheral neuropathy [22]. However, this study did not specify which variants were used to establish CYP3A4 metabolizer phenotype, and the sample size included was too small (*n* = 56) to find a large number of patients with CYP3A4 reduced function [22]. Given the large number of drugs metabolized by CYP3A4, it would be essential to investigate the impact of concomitantly administered CYP3A4 inhibitors and inducers, as well as the likelihood of phenoconversion.

### 3.2. CYP3A5

This enzyme metabolizes drugs such as cyclosporine, sirolimus, saquinavir, maraviroc, midazolam, vincristine, and statins. CYP3A5 is also associated with resistance to tyrosine kinase inhibitors and paclitaxel; however, this association has not been confirmed [38]. To date, six variants of CYP3A5 have been described (https://www.pharmvar.org/gene/CYP3A5) (accessed on 18 April 2023) [38]. The *CYP3A5**3 allele is defined by the SNP 6986A>G in intron 3, which leads to a non-functional protein in homozygous carriers [39]. Most people of European ancestry and Asian ancestry exhibit this defective gene. Therefore, between 10 and 25% of the population expresses CYP3A5, albeit this range varies greatly by ethnicity [33]. Moreover, *CYP3A5**6 and *CYP3A5**7 alleles produce a truncated protein, resulting in the absence of CYP3A5 expression [40].These are present in the European population with a frequency of 94.3% and 0.3%, respectively. When expressed, CYP3A5 can account for around 50% of the total CYP3A hepatic content, which is equivalent to CYP3A4 activity [41]. CYP3A4 and CYP3A5, on the other hand, usually share substrate specificity. Therefore, a combined phenotype including both enzymes was proposed for several drugs [33].

### 3.3. CYP2C19

This gene is highly polymorphic, with 46 alleles registered on the PharmVar platform (https://www.pharmvar.org/gene/CYP1A2) (accessed on 18 April 2023) [42]. The presence of these many polymorphisms and differences in their frequency indicates that protein expression can vary among populations. The most common polymorphism leading to a non-functional protein is *CYP2C19**2 (rs4244285, 681G>A) with a frequency of 15% in the Caucasian population. *CYP2C19**3 (rs4986893, 636G>A) is less frequent in the Caucasian population, with frequencies below 1% but it can be found in up to 6% of the Chinese population. Moreover, *CYP2C19**4 (rs28399504, 1A>G) also leads to a nonfunctional protein, but its frequency is extremely low in most populations (<0.01%). On the contrary, *CYP2C19**17 (rs12248560, -806C>T), which is found in 21% of the Caucasian population, is an enhanced-activity allele that confers the rapid metabolizer (RM) and ultra-rapid metabolizer phenotypes (UM), when present in heterozygosis or homozygosis, respectively [43,44,45]. The influence of the CYP2C19 metabolizer phenotype was investigated by Zhou et al., who described no association between CYP2C19 activity and BTZ efficacy or the incidence of peripheral neuropathy [22]. Further research is warranted.

### 3.4. CYP1A2

CYP1A2 enzyme accounts up to 16% of the total hepatic P450 pool in some individuals. Currently, there are 41 *CYP1A2* alleles described in PharmVar database (https://www.pharmvar.org/gene/CYP1A2) (accessed on 18 April 2023). The most commonly investigated alleles are *CYP1A2**1C (rs2069514), *CYP1A2**1F (rs762551), and *CYP1A2**1B (rs2470890), with frequencies in the Caucasian population of 2%, 32%, and 40%, respectively [42]. The *CYP1A2**1C allele reduces enzyme functionality, whereas the *1F and *1B alleles enhanced inducibility. *CYP1A2**1B rs2470890 has been associated with the occurrence of more serious side effects of clozapine [27]. According to our knowledge, no research has been performed to study the influence of CYP1A2 polymorphisms on BTZ efficacy and safety profile.

### 3.5. CYP2D6

CYP2D6 is expressed outside of the liver and metabolizes approximately 15–25% of the drugs used in therapeutic areas such as cardiology or oncology, although it constitutes only 2–4% of the total CYP content in the liver [46]. *CYP2D6* is highly polymorphic and exhibits a large number of functional polymorphisms that significantly alter drug metabolism. Commonly, there are different metabolizer phenotypes depending on the functionality of the alleles: poor metabolizers (PM), intermediate metabolizers (IM), normal metabolizers (NM), and ultrarapid metabolizers (UM) [16]. PMs have no enzyme activity; IMs have lower activity and UMs have increased enzyme activity compared to the NM phenotype. Usually, an increase in gene copy number or enhanced transcription is the cause of the increased activity. According to the Clinical Pharmacogenetics Implementation Consortium (CPIC) recommendation for genotype-to-phenotype translation, each allele is assigned a value between 0 and 1 to facilitate standardization [47]: UM have a diplotype with an activity score above 2.25; NM have an activity score from 1.25 to 2.25; the activity score for IM ranges from 0.25 to 1.25, and lastly, PM show an activity score of 0.

To date, more than 100 CYP2D6 alleles have been described (https://www.pharmvar.org/gene/CYP2D6) (accessed on 18 April 2023), conferring half of them a reduced activity enzyme. *CYP2D6**3 (rs35742686) and *CYP2D6**4 (rs3892097) cause a frameshift mutation and a splicing defect, respectively, that result in a truncated non-functional protein. The latter is responsible for the majority of CYP2D6 PMs found in Caucasian populations [48]. *CYP2D6**5 is effectively a whole gene deletion. This allele is present at a frequency of 1–7% across most populations and contributes to the PM phenotype [48]. The *CYP2D6**6 allele (rs5030655), similar to *CYP2D6**3, causes a truncated and non-functional CYP2D6 protein. The *CYP2D6**6 variant is more common in Caucasians, although not as frequent as *CYP2D6**4 and *CYP2D6**5 [48]. The polymorphism CYP2D6 rs5030865 defines two inactive variants, *CYP2D6**8 and *CYP2D6**14 when the reference C nucleotide is changed for an A or T, respectively. Common in Europeans, the *CYP2D6**9 rs5030656 haplotype results from the deletion of a single amino acid and is considered a reduced function allele [48]. *CYP2D6**10, defined by rs1065852 and rs1135840, is a reduced function haplotype extremely common in populations of Asian ancestry [48]. The presence of *CYP2D6**10 in homozygosity results in an IM phenotype. *CYP2D6**17 (rs28371706) was first identified in a Zimbabwean population and confers impaired CYP2D6-dependent hydroxylase activity, being highly prevalent in populations of African origin but mostly absent in others [49]. Likewise, *CYP2D6**29 (rs59421388) was originally discovered in African populations, and contributes towards a CYP2D6 IM phenotype; however, it is very rare in other populations. *CYP2D6**41 (rs28371725) is an intronic polymorphism that causes a decreased function allele that contributes towards the CYP2D6 IM phenotype. Its frequency varies among different population, but it can be considered common. Finally, *CYP2D6**7 (rs5030867), *CYP2D6**12 (rs5030862), *CYP2D6**15 (rs774671100), *CYP2D6**19 (rs72549353), *CYP2D6**56B (rs72549347), and *CYP2D6**59 (rs79292917) are extremely infrequent in most populations; however, they confer an inactive allele resulting in a non-functional protein [42,50].

The CYP2D6 gene is also variable in copy number, which in the case of an increase in more than two copies of functional alleles, confer the UM phenotype. In the case of increases in the number of copies of non-functional alleles, the functionality of the alleles would not be affected and the previously assigned IM or PM phenotype would not change.

### 3.6. CYP2C9

On the PharmVar database, 68 *CYP2C9* alleles have been registered so far [42]. The most common loss-of-function alleles are *CYP2C9**2 (rs1799853) and *CYP2C9**3 (rs1057910), which show a 12% and 7% minor allele frequency in the European population, respectively (Table 1). With this SNP *3 rs1057910, genetic changes in the locus alter the metabolism or bioactivity of therapeutic medications with nonfunctional enzymes. Leu/Leu homozygotes showed lower metabolic activity for CYP2C9 substrates, including tolbutamide and phenytoin [51]. 

*CYP2C9**2 (rs1799853) is defined by the 144Arg>Cys change in exon 3, which is related to a decrease in the function of the allele. The A allele defines the *CYP2C9**8 rs7900194 variant, which has decreased activity. In the same way, *CYP2C9* *11 rs28371685, *CYP2C9**5 (D360E, rs28371686), *CYP2C9**6, *CYP2C9**8 (R150H, rs7900194), and *CYP2C9**11 (R335W, rs28371685) variants were associated with decreased phenytoin metabolism in a black population [51]. Finally, the *CYP2C9**5 rs28371686 and *8 rs9332094 polymorphisms have been associated with decreased CYP2C9 function [42].

### 3.7. CYP17A1

The less commonly studied CYP17A1 isoform is important in diseases such as polycystic ovarian syndrome and prostate cancer. In addition, CYP17A1 catalyzes the biosynthesis of human androgens. Over 50 CYP17A1 alleles have been identified [52]. CYP17A1 is a key target for the treatment of breast and prostate cancers that proliferate in response to estrogens and androgens [52]. Steroids have been shown to affect nerve cells, and have been suggested as a therapeutic option to prevent neuropathy. *CYP17A1* rs619824 polymorphism was associated with BTZ-induced peripheral neuropathy [53]. Further research is needed to confirm this association.

### 3.8. ABCB1

BTZ is a P-glycoprotein substrate, encoded by *ABCB1,* also known as multidrug resistance protein 1 (MDR1). P-glycoprotein is an efflux pump that transports a wide range of xenobiotic compounds, ejecting them out of the tissue in which it is expressed [54]. *ABCB1* is highly polymorphic; however, three of these single nucleotide polymorphisms (rs1128503(1236C>T), rs2032582(2677G>T/A), and rs1045642(3435C>T)) have been widely studied since they affect mRNA levels, protein folding, and drug pharmacokinetics. The overexpression of *ABCB1* has been found in cell lines resistant to proteasome inhibitors. Moreover, P-glycoprotein is expressed in malignant plasma cells. In addition, the proteasome inhibitor carfilzomib is a substrate of *ABCB1*, although with BTZ, there is some controversy [55,56,57,58]. Therefore, further analyses are required to confirm whether there is an influence of *ABCB1* polymorphisms and the BTZ efficacy and safety profile, especially in patients with BTZ-induced peripheral neuropathy.

### 3.9. CIP2A

Located on chromosome 3, CIP2A is the gene encoding for the cellular inhibitor of protein phosphatase 2A (PP2A). It is an oncoprotein that inhibits PP2A and stabilizes MYC proto-oncogene in human malignancies. It also promotes anchorage-independent cell growth and tumor formation [59]. The heterozygotic and homozygotic mutations of CIP2A rs34172460 (S258A) may influence BTZ resistance in individuals with MM [60]. However, this association lacks confirmation and warrants further research.

In conclusion, *CYP3A4, CYP3A5, CYP2C19, CYP1A2, CYP2D6, CYP2C9, ABCB1*, *CIP2A,* and *CYP17A1* genes are those that are most likely to have a major influence on the metabolism and transport of BTZ, and may influence its pharmacokinetic and pharmacodynamic parameters. Table 1 shows the main polymorphisms of these genes that are of major relevance for study in the Caucasian population.

### 3.10. Other Considerations

Additionally, it is critical to identify any potential drug interactions that could result from the concurrent administration of predominantly CYP3A4 inducers and inhibitors. While the coadministration of weak CYP3A4 inducers, such as dexamethasone, does not affect the pharmacological profile of BTZ, concomitant administration of strong CYP3A4 inducers, such as rifampicin, carbamazepine, phenytoin, phenobarbital, or St. John’s wort, may result in a reduction in the clinical effect [61]. A randomized trial described that concomitant treatment of the CYP3A inhibitor ketoconazole and bortezomib led to an average increase of 35% in bortezomib exposure [62]. However, neither the proteasome inhibition nor the safety profiles were impacted by the co-administration of rifampicin, a potent CYP3A4 inducer. Moreover, the co-administration of dexamethasone, a weak CYP3A4 inducer, did not affect the exposure to bortezomib [63].

Strong and moderate CYP2D6 inhibitors have been identified, such as fluoxetine and duloxetine, respectively [61]. An open-label, crossover, pharmacokinetic drug–drug interaction study was conducted at seven institutions in the United States and Europe to determine the effect of CYP2C19 inhibitor omeprazole on the pharmacokinetics and safety profile of bortezomib in patients with advanced solid tumors, non-Hodgkin’s lymphoma, or multiple myeloma. No clear effect of omeprazole coadministration on the pharmacokinetics, pharmacodynamics, or safety profile of bortezomib was found [64]. Fluvoxamine and oral contraceptives have been found to be strong and mild inhibitors of CYP1A2, respectively. In addition, CYP1A2 has inducers such as omeprazole and smoking [61].

The significance of these inhibitors and inducers stems from their potential interaction with the activity of BTZ and their impact on the development of peripheral neuropathy and other adverse effects.

## 4. Conclusions

Currently, the use of clinical pharmacogenetics is unusual for the adjustment of BTZ chemotherapy in patients with MM, based on the scarce evidence found that was aimed at identifying polymorphisms of metabolizing enzymes and transporters as BTZ efficacy and safety biomarkers. 

However, new advances in genotyping techniques may guide targeted therapy specific to the patients’ genetic characteristics, moving forward to a Precision Medicine approach. In the long term, customized treatment strategies for MM patients may be feasible if associations between polymorphisms in enzymes and transporters and the clinical factors are verified.

## 5. Future Perspectives

Precision medicine recognizes that each patient is unique so that, through disease-specific diagnostic tests, it is possible to predict how a patient will respond to therapy, determine the best dose and/or duration of treatment. Precision medicine promises to transform the treatment of patients by offering a more predictive medicine, replacing the classic trial-and-error model. It is well-known that the pharmacokinetic and pharmacodynamic processes of drugs, and consequently the response to them, are affected by differences in the genes encoding the drug-metabolizing enzymes, the transporters that distribute drugs through the body, and the therapeutic targets.

We anticipate that this review of the available evidence will pave the way for further research into the role that enzymes and transporters play in the metabolism and transport of BTZ and how reducing the risk of adverse events, such as peripheral neuropathy, might improve clinical outcomes of BTZ-treated patients. In many circumstances, increased toxicity or low efficacy of the treatment may be avoided by establishing treatment with better specificity for each patient if such a relationship was recognized.

Moreover, the assessment of plasma drug levels would also be intriguing since therapeutic drug monitoring of BTZ could assist in identifying whether levels are within the therapeutic window or whether a dose adjustment is required. It will be possible to identify potential pharmacogenetic biomarkers for dose adjustment and/or real-world treatment recommendation in a sizable cohort of both newly diagnosed and relapsed/refractory MM patients receiving treatment with BTZ in combination with other medications, according to usual clinical practice.

Moreover, a comprehensive pharmacogenetic study evaluating BTZ responsiveness would be beneficial. A genome-wide association study (GWAS) in a large patient population would be preferred since it could lead to the identification of previously unrecognized genes and variants, associated with drug effect without requiring in-depth knowledge of disease physiology or drug pharmacokinetics or pharmacodynamics. The increased expenditures compared to a candidate-genes method implies that not all research organizations can afford to implement this technique. In the event that the latter option is used, we provided the most relevant genetic variants in Table 1 to aid in deciding which ones might be worthwhile to investigate. Lastly, to determine whether genotyping optimizes BTZ response and minimizes the likelihood of adverse events, it is important to design early genotyping procedures and demonstrate their efficacy in randomized clinical trials. Preventive testing could be applied if the pharmacogenetic correlations shown in this analysis are consolidated as reliable biomarkers. Incorporating pharmacogenetics as a variable in MM treatment recommendations may be of significant use in developing dose regimens tailored to the patient’s genotype and phenotype.

Precision medicine encompasses several areas of study, including pharmacogenetics, which need to be evaluated for its proper application. Future approaches should assess the influence of genetic variations and other disciplines such as proteomics, metabolomics, nutrigenomics, etc. A multidisciplinary approach will shed more light on therapeutic failure and differences in the incidence of serious adverse reactions.

Moreover, the potency, selectivity, pharmacokinetics, safety, and drug–drug interactions of other clinically proven proteasome inhibitors such as carfilzomib and ixazomib differ both quantitatively and qualitatively from those of bortezomib [65]. For instance, low-grade gastrointestinal toxicities were the most often noted ADRs after therapy with oprozomib (first used in humans) [66], whereas carfilzomib caused a higher prevalence of cardiovascular ADRs [67]. Regarding drug–drug interactions and the influence of CYP enzyme inhibitors and inducers, there are also differences in the various proteasome inhibitors. Strong CYP3A inhibitors have no discernible effect on the PK of ixazomib. Rifampicin, on the other hand, has demonstrated a clinically significant decrease in ixazomib exposure, supporting the avoidance of combination therapy with CYP3A inducers [68]. Based on our review of the available literature, there are no pharmacogenetic studies that analyze the influence of genetic markers on the response to proteasome inhibitors. Research is needed to further investigate the involvement of genes and possible drug–drug interactions on the pharmacokinetic, pharmacodynamic, and safety parameters of proteasome inhibitors used in the treatment of multiple myeloma.

## Figures and Tables

**Figure 1 jpm-13-00695-f001:**
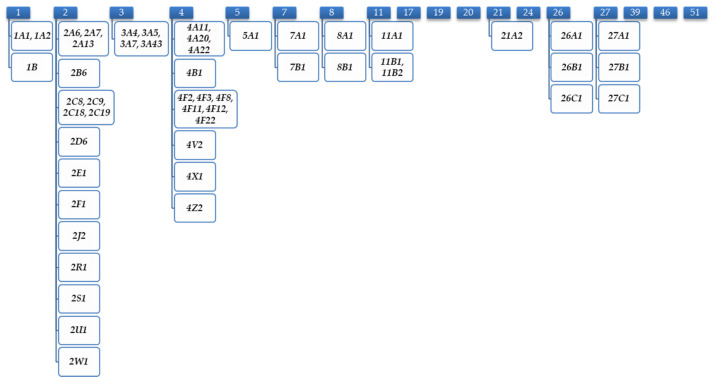
CYP pattern of cytochrome P450 hemoproteins in the human species.

**Table 1 jpm-13-00695-t001:** Summary of polymorphisms in CYP and ABCB1 genes potentially involved in BTZ pharmacokinetics and their frequency in the European population.

Gene	Polymorphism	Reference Allele	Alternative Allele	European MAF
*CYP3A4*	*3 rs4986910	A	G	0.007
*CYP3A4*	*2 rs55785340	A	G/T	0.002
*CYP3A4*	*6 rs4646438	T	TT	0
*CYP3A4*	*18 rs28371759	A	G	0
*CYP3A4*	*20 rs67666821	T	DEL	<0.1
*CYP3A4*	*22 rs35599367	G	A	0.05
*CYP3A5*	*3 rs776746	T	C	0.943
*CYP3A5*	*6 rs10264272	C	T	0.003
*CYP3A5*	*7 rs41303343	A	AA	0
*CYP2C19*	*2 rs4244285	G	A/C	0.145
*CYP2C19*	*3 rs4986893	G	A	0
*CYP2C19*	*4 rs28399504	A	G/T	0.001
*CYP2C19*	*17 rs12248560	C	A/T	0.224
*CYP1A2*	*1C rs2069514	G	A	0.02
*CYP1A2*	*1F rs762551	C	A/G	0.68
*CYP1A2*	*1B rs2470890	T	C	0.404
*CYP2D6*	*3 rs35742686	T	DEL	0.019
*CYP2D6*	*4 rs3892097	C	T	0.186
*CYP2D6*	*6 rs5030655	UN	DEL	0.02
*CYP2D6*	*7 rs5030867	T	G	0
*CYP2D6*	*8 rs5030865	C	A/G/T	0
*CYP2D6*	*9 rs5030656	CTTCT	CT (INDEL)	0.026
*CYP2D6*	*10 rs1065852	G	A/C	0.202
*CYP2D6*	*10 rs1135840	C	G	0.546
*CYP2D6*	*12 rs5030862	C	T	0
*CYP2D6*	*14 rs5030865	C	A/G/T	0
*CYP2D6*	*15 rs774671100	A	AA	0.000385
*CYP2D6*	*17 rs28371706	G	C/T	0.002
*CYP2D6*	*19 rs72549353	AGTTAG	AG	0.00015
*CYP2D6*	*29 rs59421388	C	T	0
*CYP2D6*	*41 rs28371725	C	T	0.093
*CYP2D6*	*56B rs72549347	G	A	0
*CYP2D6*	*59 rs79292917	C	T	0.002
*CYP2D6*	CNV (*5—deletion, duplication)			
*CYP2C9*	*2 rs1799853	C	T	0.124
*CYP2C9*	*3 rs1057910	A	C/G	0.073
*CYP2C9*	*5 rs28371686	C	A/G	0
*CYP2C9*	*8 rs9332094	T	C	0.001
*CYP2C9*	*8 rs7900194	G	A/C/T	0.002
*CYP2C9*	*11 rs28371685	C	T	0.002
*ABCB1*	C3435T rs1045642	A	C/G/T	0.482
*ABCB1*	C1236T rs1128503	A	G	0.584
*ABCB1*	G2677T/A rs2032582	A	CT	0.573
*CIP2A*	SNP rs34172460(S258A)	A	C	0
*CYP17A1*	rs619824	A	C	0.576

Abbreviation: *CYP2C19*: cytochrome P450 family 2 subfamily C member 19; *CYP2D6*: cytochrome P450 family 2 subfamily D member 6; *CYP3A5:* cytochrome P450 family 3 subfamily C member 5; *CYP1A2*: cytochrome P450 family 1 subfamily C member 2; *CYP2C9*: cytochrome P450 family 2 subfamily C member 9; *CYP17A1:* cytochrome P450 family 17 subfamily C member 1; *ABCB1*: ATP binding cassette subfamily B member 1; *CIP2A*: cancerous inhibitor of protein phosphatase 2A; MAF, Minor Allele Frequency.

## Data Availability

Not applicable.

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
