# Peer review of "Bortezomib Pharmacogenetic Biomarkers for the Treatment of Multiple Myeloma: Review and Future Perspectives"

_jpm, 2023, doi:10.3390/jpm13040695_

Round 1

Reviewer 1 Report

In the abstract section you summarize the purpose of their review, explaining the metabolism of bortezomib in the body, and implying that the polymorphisms in the genes encoding cytochrome p450 enzymes and p glycoprotein could explain the variable adverse effects of bortezomib. In the introduction section you explain multiple myeloma and it’s treatment landscape ( section 1.1), which is possibly not warranted for this review article, as the scope here is to explore the pharmacogenetic biomarkers of bortezomib metabolism. In section 1.2 you describe the mechanism of action of bortezomib in detail. In section 2, you describe the materials and methods. They describe the systemic review that was carried on the topic of interest, and appropriately mention the PRISMA guidelines they followed. A schematic diagram could have been included to better represent the process they followed. Also, you have not specified the time frame for which this analysis was carried out. You have introduced the abbreviation “ABCB1” without explaining /expanding the abbreviation. In section 3, you describe various enzymatic pathways that are used to metabolize bortezomib, emphasizing the role and importance of CYP3A4 , CYP2C19, CYP1A2, CYP2D6 and CYP2C9. Yiu also describe the prevalence of various polymorphisms in the Caucasian population, and potential impact on bortezomib metabolism. In an open-label, crossover, pharmacokinetic drug-drug interaction study that was conducted at seven institutions in the US and Europe between January 2005 and August 2006 to determine the effect of the cytochrome P450 2C19 inhibitor omeprazole on the pharmacokinetics and safety profile of bortezomib in patients with advanced solid tumors, non-Hodgkin's lymphoma or multiple myeloma, no clear impact on the pharmacokinetics, pharmacodynamics and safety profile of bortezomib was seen with coadministration of omeprazole was identified. (Clin Pharmacokinet. 2009;48(3):199-209) In another randomized study, concomitant administration of the CYP3A inhibitor ketoconazole with bortezomib resulted in a mean increase of 35% in bortezomib exposure (Clin Ther. 2009;31 Pt 2:2444- 58. ) Co-administration of rifampicin, a strong CYP3A4 inducer was noted to decrease the exposure to bortezomib but did not affect the proteasome inhibition or safety profiles; co-administration of dexamethasone, a weak CYP3A4 inducer did not affect the exposure to bortezomib, in a study deigned to identify the Effect of cytochrome P450 3A4 inducers on the pharmacokinetic, pharmacodynamic and safety profiles of bortezomib in patients with multiple myeloma or non-Hodgkin's lymphoma ( Clin Pharmacokinet . 2011 Dec 1;50(12):781-91). You have explored various possible mechanisms and polymorphisms that can potentially explain the variable clinical effects of bortezomib, which you state in their conclusion. It is worth exploring, especially in the current era of advanced sequencing techniques like proteomics and re-visit the topic that has been explored in the past to some degree

Author Response

Dear Editor,

Thank you for your letter and the reviewers’ comments on our manuscript entitled “Bortezomib Pharmacogenetic Biomarkers for the Treatment of Multiple Myeloma: Review and Future Perspectives” (Manuscript ID: jpm-2338095).

In this revised manuscript, we have made our best effort to address the issues raised by the reviewers. In most cases, their comments appeared to us very appropriate, and we believe that they have truly helped to improve the quality of our manuscript. All changes (highlighted in tracking mode in the MS) are detailed on a point-by-point basis in the attached separate sheets.

Response to reviewers:

In the abstract section you summarize the purpose of their review, explaining the metabolism of bortezomib in the body, and implying that the polymorphisms in the genes encoding cytochrome p450 enzymes and p glycoprotein could explain the variable adverse effects of bortezomib. In the introduction section you explain multiple myeloma and it’s treatment landscape (section 1.1), which is possibly not warranted for this review article, as the scope here is to explore the pharmacogenetic biomarkers of bortezomib metabolism.

Response:

We thank the reviewer for his/her review and comments. As the reviewer suggested, we have shorten the introduction to focus on the scope of the review. Therefore, we have deleted the following paragraphs of section 1.1.:

Since the first case was described in the 1840s, the treatment of MM has undergone a great evolution allowing patients to increase their survival to 7-8 years; even though MM remains an incurable disease. For many years, MM was treated with alkylating agents (melphalan, cyclophosphamide), doxorubicin, and corticosteroids. Then, immunomod-ulators such as thalidomide and lenalidomide and proteasome inhibitors such as borte-zomib (BTZ) appeared [8–10]. The glucocorticoid prednisone was used in MM for the first time in the 1960s, and high-dose dexamethasone was introduced in the 1980s. [11]. In addition, since the 2000s, treatments such as monoclonal antibodies (daratumumab) and CAR-T cells have been implemented, which have led to an improvement in the treatment of MM [12,13].

Even so, since it is currently an incurable disease, most patients with MM tend to relapse. Individuals who are eligible for transplant should be considered if they have never had one previously or if their first transplant resulted in a great remission length. For these patients, three combinations with daratumumab have been effective: daratu-mumab, lenalidomide, dexamethasone (DRd); daratumumab, BTZ and dexamethasone; and daratumumab, pomalidomide and dexamethasone [14].

 In section 1.2 you describe the mechanism of action of bortezomib in detail. In section 2, you describe the materials and methods. They describe the systemic review that was carried on the topic of interest, and appropriately mention the PRISMA guidelines they followed. A schematic diagram could have been included to better represent the process they followed. Also, you have not specified the time frame for which this analysis was carried out.

Response: We thank the reviewer for the suggestion. We followed PRISMA guidelines, but no diagram was made since from the 49 retrieved publications, only 2 could be finally included, since the evidence of pharmacogenetic approach of bortezomib is scarce.  Moreover, we have included the timeframe for the analysis (from bortezomib approval 2004 – to February 2023) in Methods section.

You have introduced the abbreviation “ABCB1” without explaining /expanding the abbreviation.

Response: We thank the reviewer for pointing out the mistake, it has been corrected and the complete gene name included.

In section 3, you describe various enzymatic pathways that are used to metabolize bortezomib, emphasizing the role and importance of CYP3A4 , CYP2C19, CYP1A2, CYP2D6 and CYP2C9. Yiu also describe the prevalence of various polymorphisms in the Caucasian population, and potential impact on bortezomib metabolism. In an open-label, crossover, pharmacokinetic drug-drug interaction study that was conducted at seven institutions in the US and Europe between January 2005 and August 2006 to determine the effect of the cytochrome P450 2C19 inhibitor omeprazole on the pharmacokinetics and safety profile of bortezomib in patients with advanced solid tumors, non-Hodgkin's lymphoma or multiple myeloma, no clear impact on the pharmacokinetics, pharmacodynamics and safety profile of bortezomib was seen with coadministration of omeprazole was identified. (Clin Pharmacokinet. 2009;48(3):199-209) In another randomized study, concomitant administration of the CYP3A inhibitor ketoconazole with bortezomib resulted in a mean increase of 35% in bortezomib exposure (Clin Ther. 2009;31 Pt 2:2444- 58.) Co-administration of rifampicin, a strong CYP3A4 inducer was noted to decrease the exposure to bortezomib but did not affect the proteasome inhibition or safety profiles; co-administration of dexamethasone, a weak CYP3A4 inducer did not affect the exposure to bortezomib, in a study designed to identify the Effect of cytochrome P450 3A4 inducers on the pharmacokinetic, pharmacodynamic and safety profiles of bortezomib in patients with multiple myeloma or non-Hodgkin's lymphoma ( Clin Pharmacokinet . 2011 Dec 1;50(12):781-91).

Response: We thank the reviewer for the references that allow us to improve the manuscript. We have included them all, as follows:

“An open-label, crossover, pharmacokinetic drug-drug interaction study was conducted at seven institutions in the United States and Europe to determine the effect of CYP2C19 inhibitor omeprazole on the pharmacokinetics and safety profile of bortezomib in patients with advanced solid tumors, non-Hodgkin's lymphoma, or multiple myeloma. No clear effect of omeprazole coadministration on the pharmacokinetics, pharmacodynamics, or safety profile of bortezomib was found [57].”

“A randomized trial described that concomitant treatment of the CYP3A inhibitor ketoconazole and bortezomib led to an average increase in bortezomib exposure of 35% [47]. However, neither the proteasome inhibition nor the safety profiles were impacted by the co-administration of rifampicin, a potent CYP3A4 inducer. Moreover, co-administration of dexamethasone, a weak CYP3A4 inducer, did not affect the exposure to bortezomib [48].”

You have explored various possible mechanisms and polymorphisms that can potentially explain the variable clinical effects of bortezomib, which you state in their conclusion. It is worth exploring, especially in the current era of advanced sequencing techniques like proteomics and re-visit the topic that has been explored in the past to some degree

Response:

We thank the reviewer for the suggestion. We have included the following paragraph into discussion of future perspectives “Precision medicine encompasses several areas of study, including pharmacogenetics, which need to be evaluated for its proper application. Future approaches should assess the influence of genetic variations and other disciplines such as proteomics, metabolomics, nutrigenomics, etc. A multidisciplinary approach will shed more light on therapeutic failure and differences in the incidence of serious adverse reactions.”

However, we consider the revision of proteomics techniques out of the manuscript scope.

Reviewer 2 Report

The author reviewed and discussed the pharmacogenetic status of bortezomib. 

1.  Does the pharmacogenetic status work similar in other protesomer inhibitor agents?

2. Though the genetic difference might influence on the PD and/or PK of bortezomib, the authors should have more discussion and reference of how these genetics impact on clinical treatment outcomes.

Author Response

Dear Editor,

Thank you for your letter and the reviewers’ comments on our manuscript entitled “Bortezomib Pharmacogenetic Biomarkers for the Treatment of Multiple Myeloma: Review and Future Perspectives” (Manuscript ID: jpm-2338095).

In this revised manuscript, we have made our best effort to address the issues raised by the reviewers. In most cases, their comments appeared to us very appropriate, and we believe that they have truly helped to improve the quality of our manuscript. All changes (highlighted in tracking mode in the MS) are detailed on a point-by-point basis in the attached separate sheets.

Response to reviewers:

The author reviewed and discussed the pharmacogenetic status of bortezomib. 

  1. Does the pharmacogenetic status work similar in other protesomer inhibitor agents?

Response:

We thank the reviewer for the useful question. We have added the following into discussion:

“Moreover, the potency, selectivity, pharmacokinetics, safety, and drug-drug interactions of other clinically proven proteasome inhibitors such as carfilzomib and ixazomib differ both quantitatively and qualitatively from those of bortezomib [79]. For instance, low-grade gastrointestinal toxicities were the most often noted ADRs in the therapy of oprozomib (first used in humans) [80], whereas carfilzomib caused a higher prevalence of cardiovascular ADRs [81]. Regarding drug-drug interactions and the influence of CYP enzyme inhibitors and inducers, there are also differences in the various proteasome inhibitors. Strong CYP3A inhibitors have no discernible effect on the PK of ixazomib. Rifampicin, on the other hand, has demonstrated a clinically significant decrease in ixazomib exposure, supporting the avoidance of combination therapy with CYP3A inducers [82]. Based on our review of the available literature, there are no pharmacogenetic studies that analyze the influence of genetic markers on the response to proteasome inhibitors. Research is needed to further investigate the involvement of genes and possible drug-drug interactions on the pharmacokinetic, pharmacodynamic and safety parameters of proteasome inhibitors used in the treatment of multiple myeloma.”

  1. Though the genetic difference might influence on the PD and/or PK of bortezomib, the authors should have more discussion and reference of how these genetics impact on clinical treatment outcomes.

Response:

We thank the reviewer for the suggestion. Evidence on the influence of pharmacogenes on the clinical outcomes of bortezomib is scarce, our review emphasises the importance of further research on this topic. Nevertheless, we provide relevant data on the genes of interest and their influence on functionality and possible clinical impact, in this pathology and others, when possible. In addition, we also provide background on the prevalence of serious adverse reactions, such as peripheral neuropathy.
